# De-simplifying single-tablet antiretroviral treatments for cost savings in France: From the patient perspectives to a 6-month follow-up on generics

Jean-Stephane Giraud[1]*, Melanie Doisne[1], Aurelie Chan Hew Wai[1], Catherine Majerholc[2], Erwan Fourn[2], Karine Sejean[1], Julie Trichereau[3], Brigitte Bonan[1], David Zucman[2]

1 Hospital Pharmacy, Foch Hospital, Suresnes, France, 2 HIV Department, Foch Hospital, Suresnes, France, 3 Biostatistics Department, Foch Hospital, Suresnes, France

* jsgiraud@gmail.com

**Data Availability Statement:** All relevant data are within the paper and its Supporting Information files.

## Abstract

In developed countries, most people living with HIV/AIDS are treated with costly brand single-tablet regimens. Given the economic impact, French guidelines recommend using generic antiretroviral therapy when possible to decrease antiretroviral therapy costs. We aimed to study HIV-infected patients' acceptability to switch from a brand single-tablet regimens [abacavir/lamivudine/dolutegravir (Triumeq®) or emtricitabine/tenofovir disoproxil fumarate/rilpivirine (Eviplera®)] to a treatment comprising of two pills: one is a fixed-dose generic combination of 2 Nucleoside Analogs and the second tablet is the third antiretroviral. This study was a prospective observational study in a French hospital. During their follow-up, patients on stable single-tablet regimens were made aware of the possible cost-saving. They were questioned about their willingness and barriers accepting the substitution. Participants chose between the two regimens, either to remain on single-tablet regimens or switch to the de-simplified regimen. Six months later, a second survey was given to the patient who chose to de-simplify and HIV viral load was controlled. The study included 98 patients: 60 receiving emtricitabine/tenofovir disoproxil fumarate/rilpivirine (Eviplera®) and 38 on abacavir/lamivudine/dolutegravir (Triumeq®). Forty-five patients accepted the de-simplified treatment, 37 refused and 16 were undecided and followed the decision offered by their physician. The main reason for unwillingness to switch is the number of pills (77.3%). In multivariate model analysis, male patients (p = 0.001) who have taken antiretroviral therapy for over 20 years (p = 0.04) and who retrieve their treatment in their community hospital (p = 0.03) are more likely to accept the switch. Fifty-one patients accepted to replace their single-tablet regimens and six months later, the majority was satisfied; only four returned to single-tablet regimens because of suspected side effects. Half of the people living with HIV/AIDS in our cohort accepted to switch from brand single-tablet regimens to a two-tablet regimen containing generic drugs within a process that emphasizes health expenditure savings.

**Funding:** The authors received no specific funding for this work.

**Competing interests:** The authors have declared that no competing interests exist.

## Introduction

Antiretroviral therapy (ART) has transformed the prognosis of HIV infection from a death sentence disease to a chronic disease. Nucleoside Analogs (NA) were the first antiretroviral drugs marketed in 1988. At present, the treatment for HIV infection consists of a three-drug combination antiretroviral treatment (cART) usually made up of two NA associated with a third molecule. From 1996 to 2000, early cART was made up of more than eight pills taken 2 to 3 times a day. The simplification of ART was made possible by the production of single-tablet co-formulated drugs. Recently, there has been an increase of single-tablet regimen (STR) use [1]. More and more patients are treated by STR sooner and the duration of treatment has increased. Modern STRs are far easier to take and tolerance of the drugs is nowadays excellent [2].

Thanks to ART, people living with HIV/AIDS (PLWHA) now have a near-normal life expectancy but brand STR represents a high burden for health budgets in developed countries. To allow the treatment of PLWHA in developing countries, license-free generic ART has been made available since 2000 that led to a tremendous scale-up of ART in these countries [3]. The first generic antiretroviral (ARV) drugs available in developed countries belong to the NA class. The promotion of generic ARV drugs is an opportunity for substantial cost savings [4, 5] whenever possible, as recommended by the French national guidelines [6]. In France, ART delivery is free for PLWHA. The cost is supported by Social Security (the French healthcare system). This drive to reduce public costs using generics is mainly for the payer, i.e. Social Security, given the economic impact of generics. Since 2012, the price discount for a generic drug is 60% compared with the original medication [6]. These savings will be of interest to healthcare institutions, social security, and patients even if the delivery is free.

At present, there is only one generic STR available in France which contains efavirenz, a poorly tolerated drug marketed in 1999. Therefore, many patients in developed countries receive brand STR that could be replaced by equivalent generic molecules by taking several tablets a day instead of one. This de-simplifying strategy was already studied in Canada about abacavir/lamivudine/dolutegravir (Triumeq®, Viih Healthcare) switched to abacavir/lamivudine and dolutegravir (Tivicay®, Viih Healthcare) [7, 8]. However, there are many well-recognized barriers to generic substitution: the most important being patient acceptability [9]. We, therefore, thought it was important to see if de-simplifying was possible and to better understand the reasons for acceptability or refusal of ARV generics.

We aimed to study HIV-infected patients' acceptability to switch from a brand STR to an equivalent treatment including a generic fixed-dose combination of two NA and a third antiretroviral molecule (integrase inhibitor or non-nucleoside reverse transcriptase inhibitor) in a second tablet. We decided to study patients' opinions about the process of de-simplifying single-tablet antiretroviral treatments only for cost saving before and six months after the switch.

## Methods

### Setting

We carried out a monocentric observational prospective non-randomized study. The study was conducted in a French HIV outpatient clinic between September 1st,2018 and January 31st, 2019 and concerned all PLWHA attending their semi-annual routine follow-up visit.

We chose to study two brand STRs: abacavir/lamivudine/dolutegravir (Triumeq®, Viih Healthcare) which can be switched to generic abacavir/lamivudine and dolutegravir (Tivicay®, Viih Healthcare) and emtricitabine/tenofovir disoproxil fumarate/rilpivirine (Eviplera®,

Gilead Sciences) which can be switched to generic Emtricitatine/Tenofovir disoproxil fumarate and rilpivirine (Edurant®, Janssen Cilag)

The research has been subjected to appropriate ethical review and approved by the institutional review board of the Foch Hospital following the National Commission for Data Protection and Liberties (CNIL)'s recommendations.

## Surveys and medical consultation process

Inclusion criteria were: (a) stable HIV infection with HIV viral load below the limit of detection (50 copies/mL) at the time of the visit, (b) ART including abacavir/lamivudine/dolutegravir (Triumeq®, Viih Healthcare) or emtricitabine/tenofovir disoproxil fumarate/rilpivirine (Eviplera®, Gilead Sciences), (c) absence of precariousness criteria or comprehension problems. During the visit, participants were made aware of the price difference between the two options and of the possible cost-saving. Then participants filled out a short questionnaire with the help of their physician. The survey was established by the hospital pharmacists in collaboration with the infectious disease physicians and contained four main themes: (a) socio-demographic and professional characteristics, (b) disease-related factors, (c) compliance with ART, knowledge and use of generic drugs, (d) willingness and barriers for accepting the substitution. Finally, the physician suggested several possibilities to every patient fulfilling the inclusion criteria: switching now, waiting until the next consultation, following the choice of their physician or keeping their STR. Participants chose between either regimens. For hesitant patients, a discussion was performed, and the last word was left to the patient. The physician prescribed the generic association immediately if the switch was accepted by the patient.

After the switch, patients were followed up at 3 or 6 months depending on their needs. At this follow-up visit, a second survey was given to the patients who chose to de-simplify. Patients could change their decision at any time. This survey contained five items: (a) satisfaction with the new treatment, (b) if they were bothered by having two pills to take instead of one, (c) their compliance, (d) their possible side effects, (e) if they were willing to continue on generic drugs or to go back to their previous brand. HIV viral load was controlled at this follow-up visit.

Two pharmacy residents in collaboration with the infectious disease physicians developed the two surveys proposed to participants before the potential switch and during the follow-up visit. Before their uses, these surveys were reviewed by hospital pharmacists. They were not tested on potential participants. A detailed description of the surveys is provided in S1–S3 Files.

## Data collection and analysis

Data were collected and anonymized on datasheets. Patients were informed during the admission process about the potential use of their clinical data for medical research. If they consented, then they signed a consent form authorizing the use of these data, approved following the CNIL's recommendations. This written consent form has been archived in the patient's medical record.

Prices for the six medications were obtained through the French Social Security's website. In France, chronic treatment is usually provided for one month in a single box. We used Social Security's prices to calculate the savings per month that can be generated by de-simplifying these two STR. We determined the number of patients who switched, and we estimated the monetary gain from the price difference between STR and the two substitute drugs.

There were very few missing data. These data are not specified in the following tables.

There were no sample size calculations performed before participant recruitment given the small number of PLWHA in the hospital cohort.

Basic descriptive statistics (including frequency rate, mean, standard deviation, median, percentage) were used. Furthermore, univariate and multivariate logistic regression models were used to estimate the factors associated with the unwillingness to switch to a generic association. All tests were double-sided and statistical significance was defined as a P-value < 0.05. Statistical analyses were performed with SAS® 9.4.

## Results

### Study population

We included 98 patients in the study. Patients' characteristics were representative of the population followed in our HIV department (Table 1). Most of them were male, and the mean age was 51.3 years old (standard deviation: 11.1 years old). Patients born in France were predominant (67.3%), 15 patients were originated from Sub-Saharan Africa and 17 from other

**Table 1. Sociodemographic and clinical characteristics of the 98 PLWHA who participated in our study.**

|  | N | n(%) |
|---|---|---|
| **Gender** | 98 | |
| Male | | 68 (69.4) |
| Female | | 30 (30.6) |
| **Socio-professional Status** | 97 | |
| Lower supervisory and technical occupations | | 3 (3.1) |
| Intermediate occupations (clerical, sales, service) | | 7 (7.2) |
| Lower managerial and professional occupations | | 41 (42.3) |
| Small employers and own-account workers | | 6 (6.2) |
| Higher managerial and professional occupations | | 30 (30.9) |
| Never worked or long-term unemployed | | 10 (10.3) |
| **Region of Birth** | 98 | |
| France | | 66 (67.3) |
| Maghreb | | 8 (8.2) |
| Sub-Saharan Africa | | 15 (15.3) |
| Others | | 9 (9.2) |
| **Age** | 98 | |
| Mean (Standard Deviation) | | 51.3 (11.1) |
| Median (25–75%) | | 54 (44–58) |
| **HBV Co-infection** | 97 | |
| No | | 93 (95.9) |
| Yes | | 4 (4.1) |
| **Viral Genotype determination** | 97 | |
| No | | 51 (52.6) |
| Yes | | 46 (47.4) |
| **History of genotypic resistance** | 47 | |
| No | | 37 (78.7) |
| Yes | | 10 (21.3) |
| **Time since HIV diagnosis** | 98 | |
| ≤ 10 years | | 22 (22.4) |
| 10–20 years | | 45 (45.9) |
| >20 years | | 31 (31.6) |

countries. Patients generally were employees and presented 4.1% coinfection with viral hepatitis B. The most frequent exclusion criterion was lack of understanding either due to foreign origin, poor educational level, or to psychiatric comorbidity.

In Table 2, 38 patients were on emtricitabine/tenofovir disoproxil fumarate/rilpivirine (Eviplera®, Gilead Sciences) (38.8%) and 60 on abacavir/lamivudine/dolutegravir (Triumeq®, Viih Healthcare) (61.2%). ART is mainly dispensed by community pharmacies for the PLWHA in our cohort. The majority has been used to having ART treatment: for longer than 10 years for 65 of them. They are mostly poly-medicated. Most of them (n = 85) know the right price of their treatments and are used to generic medications for comorbidities (n = 74).

In response to the generic confidence scale presented in the first survey (S1 File), patients report trusting generics medication: greatly (n = 41), (n = 41), moderately (n = 34), a little (n = 12), not at all (n = 9). Only 28 patients (29.8%) thought that taking two pills instead of one would increase the probability to forget to take one pill.

## Prescription of de-simplify ART

During the visit, 45 (46%) patients made their decision to switch to separate ART with generic drugs but one of them withdrew his consent to switch a few hours after the visit. Among the 16 undecided patients, seven followed the decision to switch proposed by their doctor and nine preferred to stay on STR (Table 3).

**Table 2. Treatments characteristics of included patients.**

| | N | n(%) |
|---|---|---|
| **Non-ARV daily comedications** | 98 | |
| No | | 35 (35.7) |
| Yes | | 63 (64.3) |
| **ART** | 98 | |
| abacavir/lamivudine/dolutegravir (Triumeq®, Viih Healthcare) | | 60 (61.2) |
| emtricitabine/tenofovir disoproxil fumarate/rilpivirine (Eviplera®, Gilead Sciences) | | 38 (38.8) |
| **ART dispensing location** | 98 | |
| Community Pharmacy | | 76 (77.6) |
| Hospital Pharmacy | | 17 (17.3) |
| Both | | 5 (5.1) |
| **Time since ART** | 95 | |
| ≤ 10 years | | 30 (31.6) |
| 10–20 years | | 45 (47.4) |
| > 20 years | | 20 (21.0) |
| **Knowledge of the price** | 98 | |
| No | | 5 (5.1) |
| Yes | | 93 (94.9) |
| **Speculated price of ART per month** | 95 | |
| <500 euros | | 3 (3.2) |
| 500–1,000 euros | | 85 (89.5) |
| 1,000–2,000 euros | | 6 (6.3) |
| >2,000 euros | | 1 (1.0) |
| **Use of generic medication for co-morbidities** | 93 | |
| Yes | | 74 (79.6) |
| No | | 19 (20.4) |

**Table 3. Carried out substitution depending on a potential agreement.**

| Patients' potential Agreement | Carried out substitution | |
|---|---|---|
| | **No** | **Yes** |
| No | 37 (100.0) | 0 (0.0) |
| Yes | 1 (2.2) | 44 (97.8) |
| Maybe | 9 (56.3) | 7 (43.7) |

Forty-seven patients remained on STR: 37 of them were opposed to the substitution, nine were reluctant. For those who declined to switch, the main reasons were related to the number of pills to take (77.3%, n = 41), the number of boxes (75.5%, n = 40) and the difficulty to take their treatment (50.9%, n = 27). There are also other reasons: patients who think they already take too many pills (35.8%, n = 19), the fear or mistrust in generic medication (30.2%, n = 16) and finally the fear of forgetting one pill (26.4%, n = 14). A few patients explained their need for discretion or their unwillingness to change their habits.

Finally, 51 (52%) patients switched to generic medications. De-simplifying concerned 15 patients on emtricitabine/tenofovir disoproxil fumarate/rilpivirine (Eviplera®, Gilead Sciences) and 36 on abacavir/lamivudine/dolutegravir (Triumeq®, Viih Healthcare) (Table 4). For our hospital, the estimated annual saving achieved was around 79,000 euros for the French Health Insurance. Switching from abacavir/lamivudine/dolutegravir (Triumeq®, Viih Healthcare) to generic abacavir/lamivudine and dolutegravir (Tivicay®, Viih Healthcare) could generate an economy of around 2,400 euros per patient per year. In the same way, switching from emtricitabine/tenofovir disoproxil fumarate/rilpivirine (Eviplera®, Gilead Sciences) to generic Emtricitatine/Tenofovir disoproxil fumarate and rilpivirine (Edurant®, Janssen Cilag) could generate an economy of around 1,200 euros per patient per year.

The categories and risk factors associated with accepting generic medications are patients older than 60 years old (p = 0.04, Table 4), male (p = 0.001), who retrieve their treatment in their community pharmacy (p = 0.04), of French nationality (p = 0.001) according to the univariate model. The multivariate model shows that male patients (p = 0.001) who have been taking ARV for over 20 years (p = 0.04) and who retrieve their treatment in their community hospital (p = 0.03) are more likely to accept the switch to generic medication.

## Six months post-de-simplification

At their follow-up visit, 47 of 51 (92%) patients in the switch group were questioned (Table 5), the survey was not proposed to four patients. One patient was excluded because the in-town pharmacist did not switch to ART generic.

During this period, only five patients (10.6%) forgot to take one of the two pills at least once. 42 patients declared never having omitted their treatment. Three patients declared having omitted only once a month.

Forty-six out of 51 (88%) patients were satisfied with the de-simplified ART and decided to continue with it.

Only four patients (8.9%) decided to return to STR. These patients had mild symptoms that they attributed to generics, such as moderate diarrhea (n = 1), abdominal pain (n = 1), bloating (n = 1), asthenia (n = 1), bad sleep (n = 1), sweat (n = 1), shortness of breath (n = 1), anorexia (n = 1). Only grade 1 adverse effects were reported. The relationship with the generic drugs was uncertain; three patients had a history of anxiety. HIV viral load remained undetectable in all patients.

In commentary and opinion columns, a few patients explained preferring STR because a single pill was easier to take and having two pills instead of one could hinder their

**Table 4. Factors associated with the unwillingness to switch to a generic association.**

| | Generic Association | | Univariate model: Factors associated with the unwillingness to switch to a generic association | | | Multivariate model: Adjusted factors associated with the unwillingness to switch to a generic association | | |
| --- | --- | --- | --- | --- | --- | --- | --- | --- |
| | Agreement | Refusal | OR* | CI95%* | p | OR* | CI95%* | p |
| | n (%) | n (%) | | | | | | |
| **Age** | | | | | 0.04 | | | |
| < 40 years old | 5 (9.8) | 9 (19.1) | 1 | - | | | | |
| [40–50[ | 11 (21.6) | 17 (36.2) | 0.6 | [0.1–2.5] | | | | |
| [50–60[ | 20 (39.2) | 15 (31.9) | 0.3 | [0.1–1.1] | | | | |
| > = 60 years old | 15 (29.4) | 6 (12.8) | **0.2** | **[0.03–0.7]** | | | | |
| **Gender** | | | | | 0.001 | | | 0.001 |
| Male | 43 (84.3) | 25 (53.2) | 1 | - | | 1 | | |
| Female | 8 (15.7) | 22 (46.8) | **4.7** | **[1.8–12.2]** | | **7** | **[2.4–20.7]** | |
| **Region of Birth** | | | | | 0.001 | | | |
| France | 44 (86.3) | 22 (46.8) | | | | | | |
| Sub-Saharan Africa | 0 (0.0) | 15 (31.9) | | | | | | |
| Others | 7 (13.7) | 10 (21.3) | | | | | | |
| **Socio-professional Status** | | | | | 0.16 | | | |
| Higher managerial and professional occupations | 19 (37.2) | 11 (23.9) | 0.5 | [0.2–1.3] | | | | |
| Others | 32 (62.8) | 35 (76.1) | 1 | - | | | | |
| **Time since HIV diagnosis** | | | | | 0.25 | | | |
| <10 years | 10 (19.6) | 12 (25.5) | 1 | | | | | |
| 10–20 years | 21 (41.2) | 24 (51.1) | 0.9 | [0.3–2.6] | | | | |
| >20 years | 20 (39.2) | 11 (23.4) | 0.5 | [0.1–1.4] | | | | |
| **Time since ART** | | | | | 0.12 | | | 0.05 |
| <10 years | 12 (24.0) | 18 (40.0) | 1 | - | | 1 | - | |
| 10–20 years | 24 (48.0) | 21 (46.7) | 0.6 | [0.3–1.5] | | 0.5 | [0.2–1.4] | |
| >20 years | 14 (28.0) | 6 (13.3) | 0.3 | [0.1–1.0] | | **0.2** | **[0.06–0.9]** | |
| **HBV Coinfection** | | | | | 0.03 | | | |
| Yes | 0 (0.0) | 4 (8.5) | | | | | | |
| No | 50 (100.0) | 43 (91.5) | | | | | | |
| **Viral Genotype** | | | | | 0.27 | | | |
| Yes | 21 (42.0) | 25 (53.2) | 1.7 | [0.7–3.5] | | | | |
| No | 29 (58.0) | 22 (46.8) | 1 | - | | | | |
| **Non-ARV daily comedications** | | | | | 0.18 | | | |
| Yes | 36 (70.6) | 27 (57.4) | 0.5 | [0.2–1.3] | | | | |
| No | 15 (29.4) | 20 (42.6) | 1 | - | | | | |
| **ART** | | | | | 0.04 | | | |
| abacavir/lamivudine/dolutegravir (Triumeq®) | 36 (70.6) | 24 (51.1) | 1 | - | | | | |
| emtricitabine/tenofovir disoproxil fumarate/rilpivirine (Eviplera®) | 15 (29.4) | 23 (48.9) | **2.3** | **[1.0–5.3]** | | | | |
| **Dispensing location** | | | | | 0.04 | | | 0.03 |
| Community Pharmacy | 45 (88.2) | 31 (66.0) | 1 | - | | 1 | - | |
| Hospital Pharmacy | 4 (7.8) | 13 (27.6) | **4.7** | **[1.4–15.8]** | | **6.4** | **[1.7–24.6]** | |
| Both | 2 (4.0) | 3 (6.4) | 2.2 | [0.3–13.8] | | 3.1 | [0.3–31.2] | |

* OR: Odds Ratios, CI95%: Confidence Interval of 95%.

**Table 5. Overall description of 47 patients with de-simplified treatment at the next follow-up visit.**

|  | n (%) |
| --- | --- |
| **Gender** |  |
| Female | 6 (12.8) |
| Male | 41 (87.2) |
| **Time since de-simplifying** |  |
| 6 months or less | 40 (85.1) |
| More than 6 months | 7 (14.9) |
| **Satisfaction** |  |
| Dissatisfied | 2 (4.2) |
| Neither Dissatisfied / Nor Satisfied | 4 (8.5) |
| Satisfied | 13 (27.7) |
| Very Satisfied | 28 (59.6) |
| **Bothered by 2 pills** |  |
| A lot | 1 (2.1) |
| Moderately | 1 (2.1) |
| A bit | 10 (21.3) |
| Not at all | 35 (74.5) |
| **Bothered by 2 boxes** |  |
| A lot | 1 (2.1) |
| Moderately | 4 (8.5) |
| A bit | 7 (14.9) |
| Not at all | 35 (74.5) |
| **Observance: omission** |  |
| Once a month | 14 (29.8) |
| Once a week | 2 (4.3) |
| Never | 31 (65.9) |
| **The omission of one of the 2 pills** |  |
| Once a month | 3 (6.4) |
| Once a week | 2 (4.2) |
| Never | 42 (89.4) |
| **Side effects** |  |
| Yes | 5 (10.6) |
| No | 42 (89.4) |
| **Return to STR** |  |
| No | 40 (85.1) |
| Yes | 4 (8.5) |
| Maybe | 3 (6.4) |
| **Increase of viral load** |  |
| Unavailable | 3 (6.4) |
| No | 44 (93.6) |

confidentiality. Most patients were happy to participate in a cost-reduction strategy and this satisfaction was the main driver for staying on de-simplified treatment.

## Discussion

About half of the PLWHA in our study accepted to de-simplify their ART to generic medications within a process that emphasizes health expenditure savings. Our results show a high level of responsibility and altruism of most of the PLWHA in our hospital, mainly older white

male patients with high socioeconomic status and stable HIV. Six months after the switch, only four patients decided to return to STR. There was no change in HIV viral load showing that the efficacy of generic ART is excellent. Cost savings were achieved thanks to an overall well-appreciated and well-tolerated approach.

In France, PLWHA take their treatment without any charge except a social contribution of 50 cents per box, as the cost is supported by Social Security. Treatment can be delivered in a hospital- or in-town pharmacy. Although there is no money exchanged for access to treatment, our study shows that most PLWHA, mainly older white male patients with high socioeconomic status and stable HIV, are aware of the cost of their ART. The mean annual ART-related cost is estimated at 15,000 euros for one patient [10]. If every patient on abacavir/lamivudine/dolutegravir (Triumeq®, Viih Healthcare) in a Canadian study (n = 607) switched to generic abacavir/lamivudine and dolutegravir (Tivicay®, Viih Healthcare), total cost would decrease by 4,325,040 dollars (3,641,683 euros) [7]. Switching from a fixed-dose combination to an efavirenz/tenofovir/lamivudine treatment with three tablets a day would save 42,500 dollars/patient (35,785 euros/patient) over a lifetime [11]. The economic argument may be compelling mainly for payers but as shown in our study most patients with high socioeconomic status, who do not pay for their ART, still know the price and are concerned about cost savings for society. Following a pandemic, such as Covid-19, health systems and the provider community will be impacted economically and financially. Can economically restrained healthcare systems handle unpredictable large-scale health crisis while remaining sustainable? Medium and longer-term planning is needed to re-balance and re-energize the economy following a crisis [12]. Medications have a high impact on health budgets. Therefore, cost savings is one possible solution to maintain patient access to their treatment, especially with chronic diseases.

A cause of non-acceptance may be a reluctance to use a generic per se [9]. Only 42% of the PLWHA in our study (n = 41) have high confidence in generic ART. This result shows however even better acceptability than the one estimated in 2013 in a French study of PLWHA showing that generic ARVs would be accepted by 44% of them but only by 17% if the pill burden was going to increase [13]. Although most physicians are confident about prescribing generic ARVs, some need more information about generic drugs [14]. Efficient information programs for patients and physicians may alleviate concerns surrounding generic substitution. ART generics are indeed prescribed and effective in most HIV patients in the world.

Our study protocol differs from the Canadian studies [7, 8] because we sought to determine the patients' perspectives on generic ARV before de-simplification and three to six months after the switch. There are only a few studies that have studied the follow-up and outcomes of patients following the de-simplification of their treatments. Also, we were able to extend this study of de-simplification to emtricitabine/tenofovir disoproxil fumarate/rilpivirine (Eviplera®). Our results are adequate concerning the acceptability of the patient [7, 9]. The better acceptability found in our study made in 2018 probably reflects an evolution of PLWHA opinion on generic ARVs over time.

Our results support previous studies: the profile of patients who agree (male, older than 65 years old and more HIV-experienced) or decline de-simplification is consistent with the literature [8]. Discretion is the main reason why patients are more likely not to de-simplify their ART: the dual location to retrieve their treatment was established because of the desire for confidentiality.

One of the factors, in our study, associated with refusal of switching the treatment is to be originated from sub-Saharan Africa, but this population is fairly under-represented in our patient sample. Nevertheless, this finding has been reported as found in other studies [15]. Several explanations are possible: first, patients from sub-Saharan Africa may be more reluctant to take generic drugs because in their country of origin, some generic drugs are of bad quality

[16] and they may also be confused with fake medications often found in local markets. Second, this illness remains a taboo subject for some communities. Due to lack of privacy, many sub-Saharan patients must hide their treatments and they often de-condition the treatment to avoid the revelation of their HIV status [7]. In this regard, STR is easier to hide. This second factor is probably predominant in our study as several of the sub-Saharan patients in the study were receiving generic drugs for other purposes than HIV.

De-simplification is not for every patient. Vulnerable populations living in unsafe environments for medication storage is a known criterion not to propose a switch [6, 7]. In France, ART like other chronic medications are delivered for a month in a single box per treatment. With this process of de-simplification, PLWHA are giving two boxes of medication for a single month. If they need discretion or with poor living conditions, this strategy can complicate proper compliance and proper medication management. Furthermore, based on their number of daily pills, poly-medicated patients may not be a choice target for this approach.

Maintaining an undetectable viral load is the main goal of HIV care today and this must not be compromised. We found no evidence that switching from STR to two tablets with one made of generic molecules impacts the suppression of viral replication [8, 17]. For patients on STR, the switch to generics increases their tablet burden and for some decreases acceptability [13, 18]. Multiple studies show better compliance with only one pill to take per day [19]. Our study shows that there was not more omission with the de-simplified treatment than with STR. For consumers, there is concern related to the tolerability of generic formulations [20]. The first generic antiretroviral drugs marketed in the developing world en 2000 consisted of molecules that had an unfavorable safety profile like stavudine. But for the generic antiretroviral drugs progressively available from 2010 in developed countries no study has shown a real tolerability issue [7]. In our study, the switch back to STR was mainly due to suspected mild side effects reported by less than 10% of patients.

In developed nations, generic antiretroviral molecules are marketed when the time of the license protections of the different drugs is over. Pharmaceutical industries are constantly researching and developing new STR and new combinations of antiretroviral molecules. They have used various strategies to delay generic competition, like longer a patent lives (e.g. co-formulating antiretroviral drugs or changing the galenic form) and may even have increased general mistrust of generics [21]. Because of license protection, only a few STR are available for now in generic form. De-simplification is likely to be only a medium-term means of cost savings while maintaining the quality of care.

Our study has limitations. Due to this study being single institutional, it is possibly not representative of patients attending other HIV centers, but the literature is consistent. Most of the switch propositions were made by a physician who had followed his patients for a long time. The results may be biased due to the good relationship between the patients and the physician. Trust is an essential part of the physician-patient relationship; it presumes mutual confidence and respect for confidentiality. The patient's full support, alongside an explanation of the issues, is necessary for this de-simplification to be successful without risking harming the patient's health [17]. We also need a longer examination time to determine if more patients wish to switch back to STR and other long-term risks. Furthermore, costs of ART vary between countries and hence the cost savings may vary slightly.

In conclusion, older white male PLWHA with high socioeconomic status and stable HIV generally accept to de-simplify their ART treatment to generic medications within an awareness process of health expenditure savings. This de-simplification does not influence viral load but must not be forced on their patients by the physicians. This approach must be implemented by health service providers to PLWHA.

## Supporting information

**S1 File. First survey.** The survey is written in French and translated in English.
(DOCX)

**S2 File. Database resulting from the first survey.**
(XLSX)

**S3 File. Second survey.** The survey is written in French and translated in English.
(DOCX)

**S4 File. Database resulting from the second survey.**
(XLSX)

## Acknowledgments

We thank all the participants, PLWHA and members of Foch Hospital. We also thank Polly Gobin for revising and editing the English version of the manuscript.

## Author Contributions

**Conceptualization:** Jean-Stephane Giraud, Aurelie Chan Hew Wai, Karine Sejean, Brigitte Bonan, David Zucman.

**Formal analysis:** Jean-Stephane Giraud, Melanie Doisne, Aurelie Chan Hew Wai, Karine Sejean, Julie Trichereau, David Zucman.

**Investigation:** Catherine Majerholc, Erwan Fourn, David Zucman.

**Supervision:** Aurelie Chan Hew Wai, Karine Sejean, Brigitte Bonan, David Zucman.

**Visualization:** Jean-Stephane Giraud, Melanie Doisne, Aurelie Chan Hew Wai, Karine Sejean, David Zucman.

**Writing – original draft:** Jean-Stephane Giraud.

**Writing – review & editing:** Jean-Stephane Giraud, Melanie Doisne, Aurelie Chan Hew Wai, Catherine Majerholc, Erwan Fourn, Karine Sejean, Julie Trichereau, Brigitte Bonan, David Zucman.

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
