## [Decision Letter · Decision Letter 0]

10 Jul 2020

PONE-D-20-14702

De-simplifying single-tablet antiretroviral treatments for cost savings in France: from the patient perspectives to a 6-month follow-up on generics

PLOS ONE

Dear Dr. Giraud,

Thank you for submitting your manuscript to PLOS ONE. After careful consideration, we feel that it has merit but does not fully meet PLOS ONE’s publication criteria as it currently stands. Therefore, we invite you to submit a revised version of the manuscript that addresses the points raised during the review process.

The reviewers both comment most specifically on the discussion section of the manuscript as being overly broad and containing statements that go well beyond the scope of the research conducted. It will be important on resubmission that you are more careful in your conclusions. 

Similarly, please pay careful attention to the precision of your language, as noted by the reviewers.

We look forward to receiving your revised manuscript.

Kind regards,

Deborah Donnell, Ph. D.

Academic Editor

PLOS ONE

Journal Requirements:

2. Please address the following:

- Please refer to any sample size calculations performed prior to participant recruitment. If these were not performed please justify the reasons.

Please refer to our statistical reporting guidelines for assistance (https://journals.plos.org/plosone/s/submission-guidelines.#loc-statistical-reporting).

- Please include additional information regarding the survey or questionnaire used in the study and ensure that you have provided sufficient details that others could replicate the analyses.

For instance, if you developed a questionnaire as part of this study and it is not under a copyright more restrictive than CC-BY, please include a copy, in both the original language and English, as Supporting Information.

In addition, please include any details of the validation of these tools, for example following pilot testing.

If performed, please provide details of the number of participants and where they were recruited from.

3. You indicated that you had ethical approval for your study and state "consent was given orally" in your submission form. Within your manuscript you also state that a consent form "was signed by the patient".

Please confirm the details regarding participant consent.

In the ethics statement in the Methods and online submission information, please ensure that you have specified (i) whether consent was informed and (ii) what type you obtained (for instance, written or verbal, and if verbal, how it was documented and witnessed).

Reviewers' comments:

Reviewer's Responses to Questions

**Comments to the Author**

1. Is the manuscript technically sound, and do the data support the conclusions?

Reviewer #1: Partly

Reviewer #2: Yes

2. Has the statistical analysis been performed appropriately and rigorously? 

Reviewer #1: Yes

Reviewer #2: Yes

3. Have the authors made all data underlying the findings in their manuscript fully available?

Reviewer #1: Yes

Reviewer #2: Yes

4. Is the manuscript presented in an intelligible fashion and written in standard English?

Reviewer #1: No

Reviewer #2: Yes

5. Review Comments to the Author

Reviewer #1: This is a study looking at patient concerns and satisfaction in desimplifying an ART regimen from 1 - 2 tablets/daily. This is important because the issue of cost savings is an important one to many healthcare settings all over the world.

MINOR COMMENTS: There are many grammatical issues particularly in the discussion.

The authors frequently state that the literature is 'coherent' . This is not the right terminology - I believe the word to use here is 'consistent'. Lines 265 - 266 "the reasson patients who retrieve ARTin the hospital is more likely not to desimplify.... " this is not grammatically correct.

Line 253 " a cause of non -acceptation..." - right word to use here is non-acceptance.

Line 296 "pharmaceutical industries developed new STR and new associations of molecules" - sentence needs to be rephrased

Use of Euros and $$ are randomly interspersed. I would suggest using one or the other or both but keep it consistent throughout the paper.

MAJOR:

My biggest concern here is that discussion of findings and the conclusions drawn from them are far more over-reaching than the scope of the study. The bulk of patients in this study were older white men with stable HIV infection with higher professional/managerial occupations - this is a big limitation. Hence line 243 "one study shows that most PLWHA are aware of the cost of their ART" is not truly accurate - it would be more appropriate to say that out of the patients in their hospital, older white male patients with higher socioeconomic status and stable HIV were more aware of the cost of ART. This is just one example of multiple similar over-reaching statements that are incorporated into the discussion. Similarly Lines 259 - 266 - the tone of this paragraph suggests that the majority of PLWH would find it acceptable to switch from STR to more complicated regimen except if they were from sub-saharan Africa. Perhaps this was not the authors' intent, but the paragraph and discussion needs to be rewritten to acknowledge the limitations in the demographics of patients surveyed.

Cost savings is a particularly relevant issue nowawayds in light of COVID-19 and worsening economic disparities due to this pandemic. As such, it would be helpful if the authors can weave that into their discussion to make it more timely and pertinent.

Reviewer #2: Reviewers Comments:

De-simplifying single-tablet antiretroviral treatments for cost savings in France: from

the patient perspectives to a 6-month follow-up on generics

Overall Comments:

This study examines the acceptability of de-simplifying STR using generic equivalents. The study, and the paper, is interesting, however, the approach and the findings are very similar to an earlier study (see references 8 & 10). The authors of this study need to state much more clearly how their study differs from this previous study, and why their study is important. More acknowledgement must be provided to the work conducted in Canada, and more strongly state that their study supports this previous work. This study in France is important and contributes to the overall discussion but it is not the first in the field so to speak and it shouldn’t present itself as such.

General Comments:

1. Introduction, Page 9, line 55 – the authors state that ‘most patients are on a STR’. Is that only in France or globally? What does ‘most’ refer to? i.e. 51%, 75% 90%? More precision is needed here as well as references for this bold statement.

2. Introduction, Page 9, line 57,58 – this statement needs to be verified by a reference.

3. Introduction, Page 9, line 65 – authors need to explain what exactly is meant by ‘the onomic impact’. The authors need to state and support if the drive to reduce costs is driven by the payer, the clinic/hospital or the individual patient?

4. Introduction – previous studies need to be mentioned here (see ref # 8, 10 for example)

5. Methods, page 11 – can the authors comment on how they addressed the possible bias on the part of the physicians to either switch or not switch regimens?

6. Methods, page 12, line 127 – could the authors explain what is meant by ‘six medications per box’? And why that is important.

7. Results, Table 1 – What demographic group does this table refer to? Only the 98 participants in the study? This needs to be clarified in the title of the table. The authors should also provide data on the comparative population who were not included in the study so the readers can see if there is any selection bias.

8. Page 15, line 156 - This sentence is somewhat misleading – only 42% trust generics ‘a lot’ [what does ‘a lot’ mean – authors need more precision here and use a different term]. This sentence needs to be restructured especially in the results section. They could use this phrase in the discussion section.

9. Page 16, line 167 – What is meant by ‘strongly opposed’? How was this determined? The term ‘strongly’ should not be used in the Results section unless it is defined somewhere.

10. Page, 16, Line 169 – why was the ‘number of boxes’ an issue? Could the authors speculate on that especially for readers outside of France who may not be familiar with this concept?

11. Page 16/17 – the discussion on cost savings need to be made clearer – especially in the Results section. The authors need to state how these savings were generated and what were they based on. Are the savings of 2,400 euros per patient per year? What percentage of the overall budget was the savings of 79,000 euros? And is that just to this one center? Please clarify these points in more detail.

12. Lines 208 to 213 appear to be exactly the same as lines 197 to 202! Please clarify this.

13. Lines 236-240 might be better suited in the introduction.

14. Page 23 – the authors should be caution not to over interpret results from the patients from sub Saharan Africa as the population in this study represents only a very small sample size. Extrapolation of this kind needs to be done with caution.

15. Page 24 – A concluding summary paragraph should be included here.

Specific Comments:

1. Abstract, Page 8, line 39/40- change …’who take antiretrovirals for 20 years’ to …’who have taken antiretrovirals…

2. Page 14, Line 149 – The sentence ‘The majority used to go… is poorly phrased and needs to be rewritten.

3. Line 223 – should it be ‘a few patients…’?

4. Page 24, line 302- should be indented as it starts a new paragraph

6. PLOS authors have the option to publish the peer review history of their article (what does this mean?). If published, this will include your full peer review and any attached files.

Reviewer #1: No

Reviewer #2: No

---

## [Author Response · Author response to Decision Letter 0]

1 Sep 2020

Responses to reviewers’ comments on the manuscript submitted by Giraud et al., “De-simplifying single-tablet antiretroviral treatments for cost savings in France: from the patient perspectives to a 6-month follow-up on generics” (Manuscript ID PONE-S-20-18418)

Paris, 23rd August 2020

Dear Editor,

We would like to thank you for your interest in our manuscript entitled “De-simplifying single-tablet antiretroviral treatments for cost savings in France: from the patient perspectives to a 6-month follow-up on generics” and for the opportunity to revise it for resubmission. We appreciate all the valuable comments from the reviewers of our work. 

We have revised our manuscript, according to the reviewers’ comments, questions, and suggestions. The resulting modifications appear with track marks in the revised manuscript. Finally, we provide answers (in blue) to all the queries.

Here are the major revisions:

- Adding the original (in French) and English translated surveys

- Nuance the discussion section

- Added a conclusion

- Reordering quotes

In case the language is still not clear enough, we would be grateful if, in the next round of revisions, you point to us the specific sentences we should improve. 

We hope that this manuscript is now suitable for publication in PLOS One.

Sincerely,

Jean-Stephane Giraud, PharmD resident

1. Please address the following:

- Please refer to any sample size calculations performed prior to participant recruitment. If these were not performed please justify the reasons. Please refer to our statistical reporting guidelines for assistance (https://journals.plos.org/plosone/s/submission-guidelines.#loc-statistical-reporting).

We appreciate your comments to improve the manuscript. Accordingly, our manuscript has been modified (lines 152 and 153): “There were no sample size calculations performed before participant recruitment given the small number of PLWHA in the hospital cohort.”

- Please include additional information regarding the survey or questionnaire used in the study and ensure that you have provided sufficient details that others could replicate the analyses. For instance, if you developed a questionnaire as part of this study and it is not under a copyright more restrictive than CC-BY, please include a copy, in both the original language and English, as Supporting Information. In addition, please include any details of the validation of these tools, for example following pilot testing. If performed, please provide details of the number of participants and where they were recruited from.

We appreciate your comments to improve the manuscript. Accordingly, our manuscript has been modified (lines 132 to 136): “Two pharmacy residents in collaboration with the infectious disease physicians developed the two surveys proposed to participants before the potential switch and during the follow-up visit. Before their uses, these surveys were reviewed by hospital pharmacists. They were not tested on potential participants. A detailed description in French and in English of the surveys is provided in Supplementary File 1 and 3.”

2. You indicated that you had ethical approval for your study and state "consent was given orally" in your submission form. Within your manuscript you also state that a consent form "was signed by the patient". Please confirm the details regarding participant consent. In the ethics statement in the Methods and online submission information, please ensure that you have specified (i) whether consent was informed and (ii) what type you obtained (for instance, written or verbal, and if verbal, how it was documented and witnessed).

We appreciate your comments to improve the manuscript. Accordingly, our manuscript has been modified (lines 139 to 143): “Patients were informed during the admission process about the potential use of their personal clinical data for medical research. If they consented, then they signed a consent form authorizing the use of these data, approved following the CNIL’s recommendations. This written consent form has been archived in the patient's medical record.”

A list of the four Supporting Information files has been added at the end of the manuscript (lines 425 to 428).

4. Reviewers' comments:

Reviewer #1: This is a study looking at patient concerns and satisfaction in desimplifying an ART regimen from 1 - 2 tablets/daily. This is important because the issue of cost savings is an important one to many healthcare settings all over the world.

MINOR COMMENTS: 

There are many grammatical issues particularly in the discussion.

The authors frequently state that the literature is ‘coherent’. This is not the right terminology - I believe the word to use here is 'consistent'. 

We apologize for this error, and we have corrected the text as suggested (lines 296 and 343).

Lines 265 - 266 "the reason patients who retrieve ARTin the hospital is more likely not to desimplify.... " this is not grammatically correct.

We apologize for this error, and we’ve changed “the reason patients who retrieve ARTin the hospital is more likely not to desimplify....” to “Discretion is the main reason why patients are more likely not to de-simplify their ART:” (lines 297 and 298)”. 

Line 253 " a cause of non -acceptation..." - right word to use here is non-acceptance.

We apologize for this error, and we have corrected the text as suggested (line 277).

Line 296 "pharmaceutical industries developed new STR and new associations of molecules" - sentence needs to be rephrased

We’ve changed “Pharmaceutical industries developed new STR and new associations of molecules” to “Pharmaceutical industries are constantly researching and developing new STR and new combinations of antiretroviral molecules.” (lines 334 and 335)”. 

Use of Euros and $$ are randomly interspersed. I would suggest using one or the other or both but keep it consistent throughout the paper.

We apologize for this error, and we have corrected the text as suggested (lines 264 and 266)

MAJOR:

My biggest concern here is that discussion of findings and the conclusions drawn from them are far more over-reaching than the scope of the study. The bulk of patients in this study were older white men with stable HIV infection with higher professional/managerial occupations - this is a big limitation. Hence line 243 "one study shows that most PLWHA are aware of the cost of their ART" is not truly accurate - it would be more appropriate to say that out of the patients in their hospital, older white male patients with higher socioeconomic status and stable HIV were more aware of the cost of ART. This is just one example of multiple similar over-reaching statements that are incorporated into the discussion. 

We appreciate your comments to improve the manuscript. Accordingly, our manuscript has been modified: 

- most of the PLWHA in our hospital, mainly older white male patients with high socioeconomic status and stable HIV (lines 250 and 251) 

- our study shows that most PLWHA in our hospital, mainly older white male patients with high socioeconomic status and stable HIV are aware of the cost of their ART (lines 259 and 260)

- most patients with high socioeconomic status, who do not pay for their ART, still know the price (line 267)

Similarly Lines 259 - 266 - the tone of this paragraph suggests that the majority of PLWH would find it acceptable to switch from STR to more complicated regimen except if they were from sub-saharan Africa. Perhaps this was not the authors' intent, but the paragraph and discussion needs to be rewritten to acknowledge the limitations in the demographics of patients surveyed.

We appreciate your comments to improve the manuscript. Accordingly, our manuscript has been modified: “The only factor, in our study, associated with refusal of switching the treatment is to be originated from sub-Saharan Africa, but this population is fairly under-represented in our patient sample. Nevertheless, this finding has been reported in as found in other studies [15].” (lines 300 to 302)

Cost savings is a particularly relevant issue nowadays in light of COVID-19 and worsening economic disparities due to this pandemic. As such, it would be helpful if the authors can weave that into their discussion to make it more timely and pertinent.

We appreciate your comments to improve the manuscript. Accordingly, our manuscript has been modified: “Following a pandemic, such as Covid-19, health systems and the provider community will be impacted economically and financially. Can economically restrained healthcare systems handle unpredictable large-scale health crisis while remaining sustainable? Medium and longer-term planning is needed to re-balance and re-energize the economy following a crisis [12]. Medications have a high impact on health budgets. Therefore, cost savings is one possible solution to maintain patient access to their treatment, especially with chronic diseases.” (lines 269 to 275)

Reviewer #2: De-simplifying single-tablet antiretroviral treatments for cost savings in France: from the patient perspectives to a 6-month follow-up on generics. Overall Comments: This study examines the acceptability of de-simplifying STR using generic equivalents. 

The study, and the paper, is interesting, however, the approach and the findings are very similar to an earlier study (see references 8 & 10). The authors of this study need to state much more clearly how their study differs from this previous study, and why their study is important. More acknowledgement must be provided to the work conducted in Canada, and more strongly state that their study supports this previous work. This study in France is important and contributes to the overall discussion but it is not the first in the field so to speak and it shouldn’t present itself as such.

We appreciate your comments to improve the manuscript. Accordingly, our manuscript has been modified: 

- In the Introduction section: “This de-simplifying strategy was already studied in Canada about abacavir/lamivudine/dolutegravir (Triumeq®, Viih Healthcare) switched to abacavir/lamivudine and dolutegravir (Tivicay®, Viih Healthcare) [7; 8]” (lines 76 to 78)

- In the Discussion section: “Our study protocol differs from the Canadian studies [7; 8] because we sought to determine the patients’ perspectives on generic ARV before de-simplification and three to six months after the switch. There are only a few studies that have studied the follow-up and outcomes of patients following the de-simplification of their treatments. Also, we were able to extend this study of de-simplification to emtricitabine/tenofovir disoproxil fumarate/rilpivirine (Eviplera®).” (lines 287 to 292)

General Comments:

1. Introduction, Page 9, line 55 – the authors state that ‘most patients are on a STR’. Is that only in France or globally? What does ‘most’ refer to? i.e. 51%, 75% 90%? More precision is needed here as well as references for this bold statement.

We did not intend to indicate “Now, most patients receive a single-tablet regimen (STR).” and we have therefore altered the text to specify that “Recently, there is an increase of single-tablet regimen (STR) use” (lines 55 and 56). We apologize for this error, and we have corrected the text. We also added a reference: [1]

2. Introduction, Page 9, line 57,58 – this statement needs to be verified by a reference.

We did not intend to indicate “Modern STRs are far easier to take and tolerance of the drugs is nowadays excellent with less than 5% ART interruption due to side-effects.” and we have therefore altered the text to specify that “Modern STRs are far easier to take and tolerance of the drugs is nowadays excellent” (lines 57 and 58). We apologize for this error, and we have corrected the text. We also added a reference: [2]

3. Introduction, Page 9, line 65 – authors need to explain what exactly is meant by ‘the onomic impact’. The authors need to state and support if the drive to reduce costs is driven by the payer, the clinic/hospital or the individual patient?

We appreciate your comments to improve the manuscript. Accordingly, our manuscript has been modified: “This drive to reduce public costs using generics is mainly for the payer, i.e. Social Security, given the economic impact of generics. Since 2012, the price discount for a generic drug is 60% compared to the original medication [4]. These savings will be of interest to healthcare institutions, social security, and patients even if the delivery is free. “(lines 68 to 72)

4. Introduction – previous studies need to be mentioned here (see ref # 8, 10 for example)

The previous studies are now referred to in line 78.

5. Methods, page 11 – can the authors comment on how they addressed the possible bias on the part of the physicians to either switch or not switch regimens?

The switch was proposed by the physician to every patient fulfilling the inclusion criteria. Patients were free to accept or deny. For hesitant patients, a gentle discussion was performed and the last word was left to the patient. This bias was mentioned in the discussion (lines 343 to 349 of the new manuscript)

Our manuscript has been modified “For hesitant patients, a discussion was performed, and the last word was left to the patient.” (lines 120 and 121).

6. Methods, page 12, line 127 – could the authors explain what is meant by ‘six medications per box’? And why that is important.

In France, chronic treatment is usually provided for one month in a single box. We used Social Security’s prices to calculate the savings per month that can be generated by de-simplifying these two STR. Our manuscript has been modified (lines 145 to 149)

7. Results, Table 1 – What demographic group does this table refer to? Only the 98 participants in the study? This needs to be clarified in the title of the table. The authors should also provide data on the comparative population who were not included in the study so the readers can see if there is any selection bias.

The title of the table has been modified to “Sociodemographic and clinical characteristics of the 98 PLWHA who participated in our study.” (lines 170 and 171)

The switch was proposed by the physician to every patient fulfilling the inclusion criteria seen during the study period. Our manuscript has been modified: “Finally, the physician suggested several possibilities to every patient fulfilling the inclusion criteria: switching now, waiting until the next consultation, following the choice of their physician or keeping their STR.” (lines 117 to 119)

The most frequent exclusion criterion was lack of understanding either due to foreign origin, poor educational level, or psychiatric comorbidity. This information was added to our manuscript (lines 167 and 168).

8. Page 15, line 156 - This sentence is somewhat misleading – only 42% trust generics ‘a lot’ [what does ‘a lot’ mean – authors need more precision here and use a different term]. This sentence needs to be restructured especially in the results section. They could use this phrase in the discussion section.

We appreciate your comments to improve the manuscript. Accordingly, our manuscript has been modified: 

- In the Results section: “In response to the generic confidence scale presented in the first survey (S1 File), patients report trusting generics medication: greatly (n = 41), moderately (n= 34), a little (n = 12), not at all (n= 9)” (lines 182 to 184)

- In the Discussion section: “Only 42% of the PLWHA in our study (n = 41) have high confidence in generic ART.” (lines 277 and 278). We moved the paragraph formerly between lines 259 and 262 to lines 278 and 281 (new manuscript).

9. Page 16, line 167 – What is meant by ‘strongly opposed’? How was this determined? The term ‘strongly’ should not be used in the Results section unless it is defined somewhere.

The term "strongly" was used to differentiate between patients who wholeheartedly agreed and patients who were reluctant to make the switch. We have corrected the text as suggested (line 194).

10. Page, 16, Line 169 – why was the ‘number of boxes’ an issue? Could the authors speculate on that especially for readers outside of France who may not be familiar with this concept?

In France, ART like other chronic medications are delivered for a month in a single box per treatment. With this process of de-simplification, PLWHA are giving two boxes of medication for a single month. If they need discretion or with poor living conditions, this strategy can complicate proper compliance and proper medication management. Our manuscript has been modified (lines 312 to 316)

11. Page 16/17 – the discussion on cost savings need to be made clearer – especially in the Results section. The authors need to state how these savings were generated and what were they based on. Are the savings of 2,400 euros per patient per year? What percentage of the overall budget was the savings of 79,000 euros? And is that just to this one center? Please clarify these points in more detail.

We have reworded the paragraph as follows: “For our hospital, the estimated annual saving achieved was around 79,000 euros for the French Health Insurance. Switching from abacavir/lamivudine/dolutegravir (Triumeq®, Viih Healthcare) to generic abacavir/lamivudine and dolutegravir (Tivicay®, Viih Healthcare) could generate an economy of around 2,400 euros per patient per year. In the same way, switching from emtricitabine/tenofovir disoproxil fumarate/rilpivirine (Eviplera®, Gilead Sciences) to generic Emtricitatine/Tenofovir disoproxil fumarate and rilpivirine (Edurant®, Janssen Cilag) could generate an economy of around 1,200 euros per patient per year”. (lines 204 to 211).

12. Lines 208 to 213 appear to be exactly the same as lines 197 to 202! Please clarify this.

We apologize for this error, and we deleted that paragraph which was a duplicate.

13. Lines 236-240 might be better suited in the introduction.

We’ve changed formerly lines 236-240 to lines 60 to 62.

14. Page 23 – the authors should be caution not to over interpret results from the patients from sub Saharan Africa as the population in this study represents only a very small sample size. Extrapolation of this kind needs to be done with caution.

As responded to reviewer n°1: We appreciate your comments to improve the manuscript. Accordingly, our manuscript has been modified: “The only factor, in our study, associated with refusal of switching the treatment is to be originated from sub-Saharan Africa, but this population is fairly under-represented in our patient sample. Nevertheless, this finding has been reported in as found in other studies [15].” (lines 300 to 302)

15. Page 24 – A concluding summary paragraph should be included here.

We have corrected the text as suggested: lines 353 to 357.

Specific Comments:

1. Abstract, Page 8, line 39/40- change …’who take antiretrovirals for 20 years’ to …’who have taken antiretrovirals…

We have corrected the text as suggested (line 40).

2. Page 14, Line 149 – The sentence ‘The majority used to go… is poorly phrased and needs to be rewritten.

We apologize for this error, and we have corrected the text as suggested: “ART is mainly dispensed by community pharmacies for the PLWHA in our cohort.” (lines 175 to 176)

3. Line 223 – should it be ‘a few patients…’?

We have corrected the text as suggested (line 242)

4. Page 24, line 302- should be indented as it starts a new paragraph

We have corrected the text as suggested (line 342)

---

## [Editor Report · Decision Letter 1]

14 Sep 2020

De-simplifying single-tablet antiretroviral treatments for cost savings in France: from the patient perspectives to a 6-month follow-up on generics

PONE-D-20-14702R1

Dear Dr. Giraud,

We’re pleased to inform you that your manuscript has been judged scientifically suitable for publication and will be formally accepted for publication once it meets all outstanding technical requirements.

Kind regards,

Deborah Donnell, Ph. D.

Academic Editor

PLOS ONE

Additional Editor Comments (optional):

Generally, the manuscript is much improved and revision has agreed to the requested changes and added needed preciseness.

However, the manuscript does need a good copy edit to correct to conventional English usage - some use of prepositions and vocabulary need correcting to standard English, and some words have incorrect use.

Two other minor details

1. I note there is still mixed use of Euro and Dollars in the discussion - if this is because of a direct extraction from a reference I would suggest a conversion in parentheses for the reader's convenience.

2. The discussion states that the only factor associated with refusal to switch treatment is origin of birth in sub-Saharan Africa (and this does not appear in the multivariate analysis) . However Table 4 has several factors that meet the p < 0.05 criteria.
---

## [Editor Report · Acceptance letter]

17 Sep 2020

PONE-D-20-14702R1 

De-simplifying single-tablet antiretroviral treatments for cost savings in France: from the patient perspectives to a 6-month follow-up on generics 

Dear Dr. Giraud:

I'm pleased to inform you that your manuscript has been deemed suitable for publication in PLOS ONE. Congratulations! Your manuscript is now with our production department. 

Kind regards, 

on behalf of

Dr. Deborah Donnell 

Academic Editor

PLOS ONE